# Perturbing Across the Feature Hierarchy to Improve Standard and Strict Blackbox Attack Transferability

**Nathan Inkawhich, Kevin J Liang, Binghui Wang, Matthew Inkawhich,**
**Lawrence Carin and Yiran Chen**
Duke University
nathan.inkawhich@duke.edu

## Abstract

We consider the blackbox transfer-based targeted adversarial attack threat model in the realm of deep neural network (DNN) image classifiers. Rather than focusing on crossing decision boundaries at the output layer of the source model, our method perturbs representations throughout the extracted feature hierarchy to resemble other classes. We design a flexible attack framework that allows for multi-layer perturbations and demonstrates state-of-the-art targeted transfer performance between ImageNet DNNs. We also show the superiority of our feature space methods under a relaxation of the common assumption that the source and target models are trained on the same dataset and label space, in some instances achieving a $10\times$ increase in targeted success rate relative to other blackbox transfer methods. Finally, we analyze why the proposed methods outperform existing attack strategies and show an extension of the method in the case when limited queries to the blackbox model are allowed.

## 1 Introduction

The adversarial machine learning community has devised many ways to cause Deep Neural Networks (DNNs) to behave unexpectedly [32, 7, 2, 21, 19]. However, the knowledge assumptions and threat models considered by an adversary are critical to attack success. In settings where access to and familiarity with the target model is restricted, there is significant room for improving adversarial methods in terms of potency and efficiency, which is the central motivation for this work.

We focus on the blackbox transfer-based adversarial threat model for DNN image classifiers. In the standard case, blackbox means the attacker does not have access to the gradients of the target model and makes no assumptions about its architecture. Transfer-based indicates that adversarial examples are created by computing adversarial perturbations using a substitute whitebox model and then attacking the target blackbox model with the resulting examples, leveraging the notion of *transferability* [23, 24, 33]. Within this threat model, our specific goal is targeted adversarial attacks, meaning the objective is to induce the target model to output a specific class.

These are significantly more challenging than un-targeted attacks, which seek to simply cause an incorrect prediction. In addition, we consider a set of more "strict" blackbox threat models in which we make varying degrees of assumptions about access to the blackbox model's training data distribution. This includes cases when the label spaces of the whitebox and blackbox models differ, and when there is *zero* training data overlap.

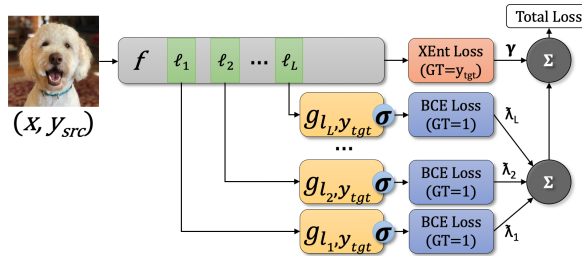

Figure 1: Visualization of forward pass construction to optimize our feature space attack objective.

A recent innovation in transfer attacks within the standard blackbox threat model is to craft perturbations based on intermediate layer representations, rather than simply optimizing the classification loss at the output layer [26, 13, 14, 18]. An example of such a method is the Feature Distribution Attack (FDA) [14], which computes adversarial examples using intermediate feature distributions at a single layer of the whitebox model. In this work, we propose to significantly improve the FDA method by extending it into a more flexible framework to allow for perturbations across the intermediate feature space, including the output layer. Our method relies on modeling the layer-wise and class-wise feature distributions of the whitebox model via auxiliary networks. We then optimize adversarial noise using the auxiliary models from across the deep feature space. The critical observation in this work is that by enforcing that a perturbed sample "looks-like" a target class sample *in multiple layers across the depth of the whitebox model*, the resulting sample is likely to have a much higher transfer rate than if we only considered a single intermediate layer or the output layer in isolation. For an intuition for how the attack works, consider Figure 1, which shows the forward pass required to generate adversarial noise given a pre-trained DNN $f$. Each $g_{\ell, y_{tgt}}$ is a feature distribution model that estimates the probability that the layer $\ell$ feature map $f_\ell(x)$ is from a sample of class $y_{tgt}$, i.e., $p(y_{tgt}|f_\ell(x))$. For a chosen $y_{tgt}$ and set of layers, we accumulate the losses w.r.t. $y_{tgt}$ at each intermediate layer and the output layer. By optimizing the sum, we are noising $x$ with $\delta$ such that $x + \delta$ lies in high probability regions of the target class at several layers across feature space. We find that this method significantly improves transferability of the generated adversarial samples.

To evaluate our methods, we consider both standard and strict blackbox transfer cases. In the standard case, our attacks show state-of-the-art targeted transfer performance between popular ImageNet-1K [4] DNN models. In some cases, we improve the targeted success rate to over $55\%$, an absolute increase of about $50\%$ over traditional output-layer-based methods. In the strict blackbox scenario, we evaluate our method under three separate relaxations of the common transfer assumption that the source and target models are trained on the same data distribution and share a label space. The results show that our feature-based attacks are significantly more potent than attacks generated at the output layer in all three situations. In an effort to explain why our methods yield high transferability, we also analyze the effects of our attacks and show that they cause significantly more disruption in the intermediate space than competing methods. Finally, we show that the noise generated with our methods provides a more useful prior direction for query-based attacking methods that incorporate prior information. With relatively few queries to the blackbox model, our method can be enhanced to yield over $90\%$ targeted success rates.

## 2 Related work

In transfer-based blackbox adversarial attack research, many of the most popular methods build directly upon whitebox attacks [7, 15, 19], adding optimization tricks, regularization, and ensembling of whitebox models to boost transferability [15, 23, 5, 38, 17, 37]. The primary optimization goal of such methods is to cause the adversarial sample to cross the whitebox model's decision boundary through consideration of the classification loss. For example, Dong et al. [5] includes a momentum term in the optimization step of [15]; Wu et al. [36] weights the gradients through the residual connections and blocks differently; Liu et al. [17] and Tramèr et al. [34] use an ensemble of whitebox models; Xie et al. [37] use diverse inputs for more robust noise; and Sharma et al. [28] and Dong et al. [6] use low frequency constraints and translation-invariance respectively to generate un-targeted attacks for defended DNNs. These methods show reasonable un-targeted performance, but very limited targeted performance at scale, if any ([6, 34, 36, 37] only consider un-targeted attacks).

There are also some recent methods that focus on improving transfer attacks by considering the feature space of the source model, to develop noise that is less overfit to the generating architecture. Zhou et al. [38] develop a regularizer for a traditional un-targeted attack that encourages adversarial examples to have significantly different intermediate feature representations. Huang et al. [11] adjusts an existing un-targeted adversarial example to have a larger effect in a pre-specified layer of feature space. Rozsa et al. [26] and Inkawhich et al. [13] develop targeted transfer attacks in feature space, representing the "target" as a single point in a single layer of the feature space. The objective functions then minimize the $L_2$ distance between a source and target point in the feature space. With this limited modeling of the target class, scalability to larger models and datasets proves difficult, and success is sensitive to the choice of the target sample. To improve targeted transfer performance at scale, Inkawhich et al. [14] explicitly model the class-wise feature distributions at multiple layers of the whitebox model, so as to have a more descriptive representation of the target class when attacking. However, the method only considers attacking from a single layer (with no options for extensions),

and the primary result is that some individual layers are better than others to transfer from. As is common in all of the aforementioned methods, results are only shown for transfer scenarios where the whitebox and blackbox models are trained on the same dataset. For comparison, we develop a flexible framework that allows for perturbations of multiple layers simultaneously, evaluate in novel cross-distribution settings, and show an integration of the method with a query-based attack.

## 3   Methodology

**Notation.** We follow the notation conventions of Inkawhich et al. [14] for feature distribution modeling. To capture the layer-wise and class-wise feature distributions of a given pre-trained DNN $f$, we first specify a set of layers $\mathcal{L} = \{\ell_1, ..., \ell_J\}$ and classes $\mathcal{C} = \{c_1, ..., c_K\}$ of interest. Using the training dataset of $f$, we train binary classification auxiliary models on the feature maps at the layers in $\mathcal{L}$ for the classes in $\mathcal{C}$. Importantly, all weights of $f$ stay fixed throughout this process. Let $f_\ell(x)$ be the layer $\ell$ feature map of $f$ for input image $x$. Then, $g_{\ell,y}$ is a binary auxiliary model that inputs $f_\ell(x)$, and outputs the probability that the feature map is from an input of class $y$ (i.e., $g_{\ell,y}(f_\ell(x)) = p(y|f_\ell(x))$). Note, the training of the necessary auxiliary models is done prior to the attacking process, and requires only a pre-trained model $f$ and its corresponding training dataset. It is also worth noting that each auxiliary model is small and individually very inexpensive to train (see Appendix B). Finally, let $f(x)$ be the predicted probability distribution over the set of classes on which $f$ was trained, and $F(x) = \text{argmax } f(x)$ be the classification prediction.

**Preliminaries.** Once trained, we can use the modeled feature distributions to attack. Given our interest in targeted attacks, we start from a source image $x$ initially classified as $y_{src}$ (i.e., $F(x) = y_{src}$) and generate a perturbation $\delta$ such that $F(x + \delta) = y_{tgt}$ for a pre-specified target class $y_{tgt}$, with $y_{src} \neq y_{tgt}$. To keep the noise approximately imperceptible and thus adversarial, for the following objectives we constrain the $L_p$ norm of $\delta$, such that $||\delta||_p \leq \epsilon$, and enforce that the perturbed input's pixel values exist in $[0, 1]$.

**FDA.** Our proposed attack framework builds on *FDA+fd*, the top performing variant of [14], which is noted here as *FDA* and described as

$$\max_\delta p(y = y_{tgt}|f_\ell(x + \delta)) + \eta \frac{||f_\ell(x + \delta) - f_\ell(x)||_2}{||f_\ell(x)||_2}. \tag{1}$$

This method has two main components, both of which are optimized at a *single* pre-specified layer $\ell$. The first component is the feature distribution piece, which takes the form of $p(y = y_{tgt}|f_\ell(x + \delta))$. By maximizing this probability through optimizing $\delta$, we are perturbing the input image such that the layer $\ell$ feature map $f_\ell(x + \delta)$ lies in a high probability region of the target class in feature space. In other words, the perturbed feature map resembles a feature map that will be classified as the target class. The second component is the feature disruption term, which enforces that the feature map of the perturbed input is significantly different than the feature map of the original input.

**FDA+xent.** A key contribution in this work is the extension of (1) to include *multi-layer* information. One way to do so is to incorporate the whitebox model's output prediction as part of the attack objective. Let $H(f(x), y)$ be the standard cross-entropy loss between the predicted probability distribution $f(x)$ and the target distribution $y$ (commonly a one-hot distribution). We include this term in the *FDA* objective to create *FDA+xent*, as described by

$$\max_\delta p(y = y_{tgt}|f_\ell(x + \delta)) + \eta \frac{||f_\ell(x + \delta) - f_\ell(x)||_2}{||f_\ell(x)||_2} - \gamma H(f(x + \delta), y_{tgt}), \tag{2}$$

where $\gamma > 0$, weighting the contribution of cross-entropy term. This attack objective optimizes the noise such that the layer $\ell$ feature map is in a high-probability region of the target class, and that the output prediction of the whitebox model is of the target class. Addition of the cross-entropy term is motivated by the correlation results from Inkawhich et al. [14], which show that the probability distribution over the classes, as measured in the "optimal transfer layer," has low correlation with the probability distribution over the classes at the output layer. This indicates that the vanilla *FDA* in (1) does not reliably create targeted examples for the whitebox model, which may limit the effectiveness of the attack; we explore this further in Section 4.1.4.

**FDA$^{(N)}$.** To further generalize the multi-layer framework, we extend the objective function to allow for optimization in *multiple intermediate layers*. We call this new attack *FDA$^{(N)}$*, with objective:

$$\max_\delta \sum_{\ell \in \mathcal{L}} \lambda_\ell \left[ p(y = y_{tgt}|f_\ell(x + \delta)) + \eta \frac{||f_\ell(x + \delta) - f_\ell(x)||_2}{||f_\ell(x)||_2} \right], \tag{3}$$

where $\sum_{\ell \in \mathcal{L}} \lambda_\ell = 1$, $\lambda_\ell > 0$, and the $N$ in the name refers to the number of layers in $\mathcal{L}$. Note that for $N = 1$, the attack objective is the same as (1). Both the *+xent* and *multi-intermediate-layer* extensions significantly increase the capability and flexibility of the method, while maintaining the intuition of attacking primarily based on feature space information. These two extensions can also be straightforwardly composed to form a $FDA^{(N)}$+*xent* attack.

**Optimization.** The final step in the methodology is the optimization of the above objective functions. Practically, we leverage the autograd feature of PyTorch by assembling the forward pass of the attack as shown in Figure 1. This step involves connecting the required auxiliary models to the corresponding layers of $f$ and computing the final loss with proper weighting. We then use an iterative projected gradient descent with momentum procedure [5] to apply the adversarial noise to the input while maintaining the norm and natural image constraints.

## 4    Experiments

We evaluate the multi-layer *FDA* framework in a variety of settings. In Section 4.1, we consider the standard blackbox transfer case, where knowledge of the target model's training dataset is assumed, and test transferability between ImageNet-1K models. We examine single-source single-target model transfers, ensemble transfers, distal adversaries, and analyze why and how the attacks work through the lens of feature disruption. In Section 4.2 we evaluate transferability in cross-distributional settings, where the source and target model are trained on different data distributions and label spaces. Finally, in Section 4.3 we show an extension of the method when limited queries to the target model are allowed during attack generation.

### 4.1    ImageNet transfer experiments

**Experimental Setup.** In our analysis, we use the following ImageNet-1K models from the PyTorch Model Zoo: ResNet-34/50/101/152 (RN34, RN50, RN101, RN152) [9]; VGG16bn and VGG19bn [30]; MobileNetv2 (MNv2) [27]; and DenseNet-121/161/201 (DN121, DN161, DN201) [10]. The notation RN50→DN121 means attacks are generated on a RN50 whitebox (source) model and transferred to a DN121 blackbox (target) model. The primary metrics of attack success are the error rate and targeted success rate (tSuc) in the blackbox model, where all clean samples are correctly classified by both the source and target models. For the attack settings, we use the common $L_\infty$ $\epsilon = 16/255$ and $\text{perturb\_iters} = 10$ (see Appendix F for perturbed samples at this $\epsilon$). The architecture of each auxiliary model $g_{\ell,y}$ is a simple Conv-Conv-FC scheme, which is significantly more parameter efficient than the FC-FC in [14].

To find the best combinations of layers for $FDA^{(N)}$ attacks, we use an iterative greedy search on a held out part of the ImageNet validation set and find value in including up to 5 layers. Crucially, feature space attacks have been shown to be *blackbox model agnostic*: the optimal transfer layer for a whitebox model does not change for different target blackbox models [13, 14]. For this reason, we find the optimal layer sets for the RN50 and DN121 whitebox models only once, in the RN50→DN121 and DN121→RN50 transfer scenarios, respectively. When attacking any other blackbox model, we use the previously found layer combinations. Practically, an attacker may have two models in a sandbox environment, where one is treated as a whitebox and the other a blackbox. They may then find the optimal layer settings in this sandbox environment, which can be used to attack any other blackbox of interest. The layer decoding scheme and the sequence in which the layers are added to the attack is shown in Appendix A. Finally, we weight all intermediate layers equally, i.e. $\lambda_\ell = 1/N$. Additional details regarding experimental setup are in Appendix B.

#### 4.1.1    Transfer results in standard blackbox settings

We perform single-source single-target model transfers for RN50→{DN121, VGG16bn, RN152, MNv2} and DN121→{RN50, VGG16bn, DN201, MNv2}. For baselines, we compare against the popular TMIM [5] and TMIM+SGM [36], both of which search for adversarial directions using *only output layer information* of the whitebox model. $FDA^{(1)}$ [14] is also considered a baseline. In an effort to emphasize the transferability of the noise generation technique when the blackbox model architecture may be unknown, we favor settings where the source and target model are from different architectural families to avoid potential biases.

The procedure for measuring the transfer results in Table 1 is as follows. For each source-target model pair, we randomly select 15,000 source images from the ImageNet validation set for which both models initially predict the correct source label. For each source sample, we then randomly

Table 1: Full-ImageNet Transfer Results (notation = error / tSuc, $\epsilon = 16/255$)

| Attack \\ Target | Whitebox Model = RN50 | | | | Whitebox Model = DN121 | | | |
|---|---|---|---|---|---|---|---|---|
| | DN121 | VGG16bn | RN152 | MNv2 | RN50 | VGG16bn | DN201 | MNv2 |
| TMIM | 44.7 / 3.0 | 48.7 / 1.5 | 41.4 / 3.2 | 54.9 / 0.1 | 46.2 / 2.3 | 49.6 / 1.3 | 44.6 / 5.5 | 58.5 / 1.0 |
| TMIM+SGM | 47.4 / 4.5 | 51.9 / 2.4 | 43.1 / 4.2 | 61.1 / 2.2 | 54.8 / 4.7 | 53.8 / 2.8 | 51.4 / 10.0 | 66.1 / 3.2 |
| $FDA^{(1)}$ | 90.3 / 19.4 | 86.6 / 13.6 | 90.8 / 17.1 | 84.7 / 7.1 | 91.8 / 20.4 | 90.6 / 20.6 | 94.8 / 35.3 | 88.4 / 10.7 |
| $FDA^{(2)}$ | 93.1 / 28.4 | 91.5 / 22.9 | 92.7 / 22.0 | 89.8 / 10.7 | **95.1** / 22.3 | **94.7** / 26.3 | 96.6 / 38.9 | **93.0** / 12.4 |
| $FDA^{(3)}$ | 94.1 / 31.0 | **93.0** / 26.1 | 93.2 / 22.9 | **91.0** / 11.8 | 95.1 / 22.2 | 94.7 / 26.0 | **96.7** / 37.1 | 92.8 / 12.6 |
| $FDA^{(4)}$ | **94.2** / 38.1 | 92.4 / 30.7 | **93.6** / 30.6 | 90.2 / 14.9 | 92.9 / 43.7 | 92.4 / 42.6 | 96.3 / 67.5 | 88.9 / 20.9 |
| $FDA^{(5)}$ | 94.2 / 38.6 | 92.7 / 31.0 | 93.5 / 29.7 | 90.9 / 15.1 | 93.8 / 44.4 | 93.4 / 44.4 | 96.6 / 68.2 | 90.6 / 22.4 |
| $FDA^{(1)}$+xent | 84.4 / 40.3 | 80.9 / 24.9 | 85.0 / 42.3 | 78.6 / 13.2 | 88.8 / 37.4 | 87.8 / 32.7 | 93.3 / 65.3 | 85.1 / 16.9 |
| $FDA^{(2)}$+xent | 89.0 / 51.7 | 86.8 / 37.2 | 88.2 / 48.1 | 84.3 / 18.7 | 92.7 / 40.5 | 92.5 / 40.5 | 95.5 / 70.3 | 90.1 / 19.4 |
| $FDA^{(3)}$+xent | 90.2 / 55.2 | 88.3 / 41.6 | 89.2 / 49.2 | 85.6 / 21.1 | 92.9 / 41.7 | 92.5 / 41.8 | 95.7 / 70.6 | 90.0 / 20.4 |
| $FDA^{(4)}$+xent | 90.5 / 57.3 | 88.2 / 42.8 | 89.7 / **53.2** | 85.4 / 22.4 | 91.3 / 49.4 | 90.9 / 46.2 | 95.3 / 76.5 | 87.1 / 22.6 |
| $FDA^{(5)}$+xent | 90.9 / **57.9** | 88.8 / **43.5** | 89.7 / 51.6 | 86.4 / **22.9** | 92.2 / **50.1** | 92.1 / **48.0** | 95.6 / **77.1** | 88.8 / **24.4** |

select 5 target labels from the ImageNet label set, and execute a targeted attack towards each. Thus, every table entry is an average computed over 75,000 targeted attacks. Both error rate and targeted success rate, as measured in the blackbox model, are reported as "error / tSuc".

Our first major observation is that the *+xent* component alone gives large gains in tSuc. If we compare $FDA^{(1)}$ to $FDA^{(1)}$+*xent*, in several cases it leads to a tSuc increase of over $20\%$; for example, in RN50→DN121 tSuc increases from $19.4\%$ to $40.3\%$. Further, we observe +*xent* consistently helps, regardless of how many layers are used. The largest performance gain observed by +*xent* is in the DN121→DN201 case, where in the 1 attack layer case, tSuc increases by $30\%$. The next result of interest is the performance gain by adding multiple intermediate layers ($FDA^{(N)}$, $N > 1$). As $N$ is increased, tSuc nearly doubles in all cases. For perspective, when we compare these results to the baseline *TMIM*-based methods, the targeted success rate in many cases improves by more than $10\times$.

We also observe that *+xent* and *multi-intermediate-layer* are complementary components: in all transfer scenarios the most powerful targeted attacks arise from using both. Specifically, consider the $FDA^{(5)}$+*xent* attack, which is the most effective in all but one case. When VGG16bn is the target model, on average this attack outperforms *TMIM+SGM* by $37\%$ error / $43\%$ tSuc and $FDA^{(1)}$ by about $2\%$ error / $29\%$ tSuc. An interesting observation is that the +*xent* term harms error performance, albeit less relevant because our focus is targeted attacks, and the methods have been optimized for tSuc. However, in all cases the best $FDA^{(N)}$ method nearly doubles the error induced by *TMIM+SGM*.

### 4.1.2 Comparison to ensemble methods

Ensemble-based approaches [17] form a separate family of methods to generate transferable adversarial examples, deviating from our previous single-source model assumption. In this setting, the attacker trains an ensemble of models (with whitebox access to each) and generates noise based on the output of the ensemble with traditional attacking techniques, e.g., *TMIM*. A natural iterate in ensemble methods is growing the numbers of models. We compare the effect of adding a layer within the $FDA^{(N)}$+*xent* framework versus adding a model to the generating ensemble using the same attack settings as

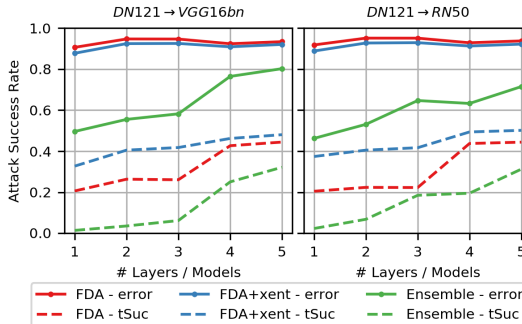

Figure 2: FDA versus ensemble attacks.

in the previous experiment. To avoid conflicts with models already in use, in every transfer case we use the following sequence for adding models to the ensemble: [whitebox, DN161, RN101, VGG19bn, RN34]. For example, in the RN50→DN121 transfer case, using a 3-model ensemble means generating noise from RN50, DN161, and RN101.

The results of these experiments are shown in Figure 2, with extensions in Appendix C. The *FDA* methods hold a wide margin over ensemble methods in almost all cases for both error and tSuc, especially $FDA^{(N)}$+*xent*. Also, although adding models to the whitebox ensemble tends to increase attack performance (as expected), large jumps in the ensemble's performance usually occur when a model from the same family as the blackbox model is added. For example, in the DN121→VGG16bn

case, a performance spike happens when the fourth model gets added, which is the VGG19bn model. Similarly, in the DN121→RN50 case, the ensemble method has the greatest increases when the third and fifth models are included, which are the RN101 and RN34 models, respectively.

### 4.1.3 Distal transfers

Distal adversarial examples are generated by starting from random noise and optimizing for high prediction probability for some target class [22]. Transferring distals between ImageNet models in the standard blackbox setting can make for an interesting test of transferability. Although distals are not our main focus, they provide a compelling future direction to study, as they depend less on the source image/class than standard transfers. This eliminates one axis of variability that may better isolate the quality of the

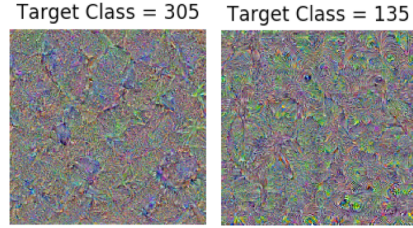

Figure 3: Samples of distals.

attacking algorithm. For these tests, we do not constrain $\epsilon$ and increase the perturbation budget to allow each attack to perturb for 200 iterations.

The results are shown as top-1 / top-5 tSuc rates, as averaged over 4,000 distals, each optimized towards a random target class. We use *TPGD* [19] as our baseline instead of *TMIM*, as momentum in the optimizer empirically harms performance in distal generation. Table 2 shows that results of the distal transfers align with the previous trends, and sample distals are shown in

Table 2: Distal transfer tSuc rates (top1 / top5)

| attack | RN50→DN121 | RN50→VGG16bn |
|---|---|---|
| TPGD | 10.1 / 21.0 | 6.1 / 13.4 |
| $FDA^{(1)}$+xent | 46.9 / 68.1 | 30.6 / 50.6 |
| $FDA^{(2)}$+xent | 58.6 / 78.8 | 38.0 / 62.5 |
| $FDA^{(3)}$+xent | 64.0 / 83.6 | 44.6 / 69.9 |
| $FDA^{(4)}$+xent | **65.8 / 85.3** | **48.1 / 73.0** |

Figure 3. Distals generated with the $FDA^{(N)}$+*xent* method are significantly more transferable. On average, the $FDA^{(4)}$+*xent* causes increases of $48\%$ top-1 / $62\%$ top-5 tSuc over the baseline method.

### 4.1.4 Analysis of *multi-intermediate-layer* and *+xent*

To analyze the effects of the *multi-intermediate-layer* and *+xent* components, we perform a feature disruption analysis [14]. Disruption measures the effect of an adversarial perturbation on the feature space of a model, w.r.t. the target class. In this setting, we measure the disruption caused by a targeted attack, at layer $\ell$ of model $f$ as: $\text{disruption}_\ell = p(y = y_{tgt}|f_\ell(x + \delta)) - p(y = y_{tgt}|f_\ell(x))$. $f$ can be either a whitebox or blackbox model and the attack that generates $\delta$ may be any method. Figure 4 shows the disruption caused by several attacks for RN50→DN121 (see Appendix D for more). Recall, the decoding for the layer notation is shown in Appendix A. For example, the $FDA^{(1)}$ attack uses layer 10 of the *whitebox* model, and $FDA^{(4)}$ uses layers 10, 5, 7, and 11. These attacking layers directly correspond to the layers on the x-axis of the whitebox subplot (but not the blackbox subplot). Any disruption caused in the blackbox model is not specifically optimized for by a $FDA^{(N)}$ attack, rather the disruption is a result of the perturbations crafted using the whitebox model's layers.

Firstly, the *TMIM* attack is shown to have drastically different effects in the whitebox and blackbox models. Since the noise is generated specifically for the output layer of the whitebox, it only causes high disruption in the last few layers of the whitebox, and very little disruption in the blackbox. For the $FDA^{(1)}$ method, we already see a significantly different pattern, namely a much higher disruption in the intermediate layers of both models. Now, consider the effects of adding multiple intermediate layers. As we look from $FDA^{(1)}$ to $FDA^{(2)}$ to $FDA^{(4)}$, the disruption is noticeably increased in the earlier and later layers, which is expected as the added intermediate layers are both earlier and later than the first layer. Finally, consider the *+xent* component. Without it, the $FDA^{(N)}$ methods have

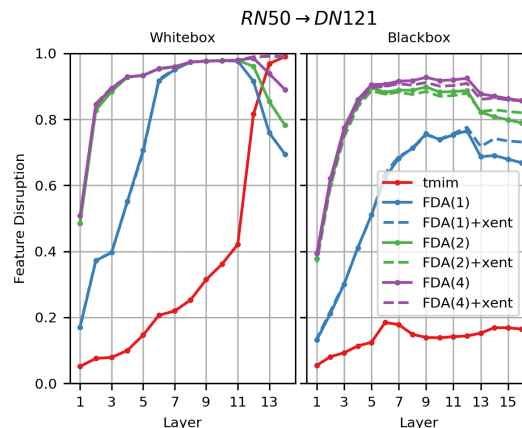

Figure 4: Disruption of features caused by attacks.

sharp decreases in the last few layers, despite high intermediate disruption. The inclusion of this term significantly boosts disruption in the final layers of both the whitebox and blackbox. Given the large

effects of our feature space methods throughout the models, we say that the methods attack along the feature hierarchy.

## 4.2 Cross-distribution experiments

An often overlooked (but critical) detail in standard blackbox transfer attacks is the underlying assumption that the whitebox and blackbox models are trained on the exact same dataset [5, 17, 36, 13, 14, 19, 33]. This implies that the adversary can find such a pre-trained model in the open-source domain or actually has the blackbox model's dataset and can train their own whitebox model. In many cases, the dataset used to train the targeted model may not be publicly available due to proprietary [8, 31], security [16, 29], or privacy [1, 25] issues. Making too strong of an assumption about the availability of the target model's data may lead to overly optimistic transfer results. Here, we evaluate the performance of transfer attacks under several more "strict" blackbox threat models, where the whitebox model is not trained on the exact same data distribution or label space as the blackbox model. We hypothesize that it is more challenging to create transferable adversarial examples if the source and target models are not trained on identical data.

We are primarily concerned with three "cross-distribution" transfer scenarios:
1. *No training data overlap between the whitebox and blackbox, but significant label space overlap.*
2. *The blackbox is trained on a subset of the whitebox's training dataset and label space.*
3. *The whitebox is trained on a subset of the blackbox's training dataset and label space.*

These three cases represent several common situations that may be encountered in reality. For example, scenarios 1 and 3 may arise when the attacker is generally aware of the types of classes that the target model is trained on, but has to go out and collect their own data for training the source model. Scenario 2 may arise when the attacker has access to a large model with a high variety of classes, and wishes to attack a smaller target model designed for a subset of the classes.

**Experimental setup.** To accommodate these scenarios, we create three new "Restricted-ImageNet" datasets (RINet), which are subsets of ImageNet-1k (Full-INet). We design *RINet-A* and *RINet-B* to each be 15-class datasets, sharing 10 label classes while each having 5 unique label classes. Each of the classes in *RINet-A* and *RINet-B* consist of WordNet [20] super-classes aggregating a related subset of ImageNet-1K classes. Critically, the classes chosen to comprise each of the shared classes of *RINet-A* and *RINet-B* are disjoint. For example, the "bird" class of RINet-A is composed of the ImageNet-1k classes: [10:'brambling', 13:'junco', 16:'bulbul', 17:'jay', 18:'magpie'], while RINet-B contains [12:'housefinch', 14:'indigobunting', 15:'robin', 19:'chickadee', 20:'waterouzel']. As a result, *RINet-A* and *RINet-B* have *zero training data overlap*. The *RINet-C* dataset is comprised of 20 classes and is meant to be a bigger subset of Full-INet than RINet-A and RINet-B. We train RN50 and DN121 models on each of the RINet datasets, then train the auxiliary models necessary to attack with the $FDA^{(N)}$+*xent* framework. For more details regarding setup and the dataset splits, see Appendix E. Hereafter, the notation RN50-A indicates the RN50 model trained on RINet-A.

**Scenario 1.** Table 3 shows the error / tSuc results for all three cross-distribution scenarios. First, consider the no training data overlap case as represented by RN50-A→DN121-B and RN50-B→DN121-A. For baselines, we include the RN50-A→DN121-A and RN50-B→DN121-B. As hypothesized, the cross-distribution transfer performance is worse than the within-distribution performance. However, the *FDA*-based methods still perform well, and significantly better than *TMIM*. For the DN121-B

Table 3: Transfer results for cross-distribution tests (notation = error / tSuc)

| Attack \\ Source | Target Model = DN121-B | | | Target Model = DN121-A | | | Target Model = DN121-Full | | |
|---|---|---|---|---|---|---|---|---|---|
| | RN50-A | RN50-B | RN50-Full | RN50-A | RN50-B | RN50-Full | RN50-A | RN50-B | RN50-C |
| TMIM | 37.8 / 11.7 | 33.5 / 21.4 | 16.4 / 3.3 | 29.9 / 19.2 | 34.8 / 11.4 | 12.6 / 2.7 | 29.2 / 0.9 | 32.0 / 1.0 | 35.3 / 2.7 |
| TMIM+SGM | 43.0 / 16.6 | 38.6 / 27.8 | 19.2 / 5.2 | 36.6 / 27.0 | 38.3 / 14.6 | 16.2 / 4.6 | 30.4 / 1.2 | 32.9 / 1.4 | 37.8 / 3.3 |
| FDA$^{(1)}$ | 67.7 / 45.8 | 70.9 / 60.8 | 38.6 / 21.0 | 66.0 / 57.3 | 67.0 / 40.8 | 34.8 / 18.9 | 53.8 / 6.9 | 58.1 / 8.1 | 75.8 / 14.1 |
| FDA$^{(2)}$ | 72.3 / 54.1 | 75.5 / 68.0 | 53.9 / 37.1 | 72.5 / 65.9 | 71.3 / 48.4 | 52.4 / 36.9 | 59.3 / 9.4 | 61.5 / 9.2 | 80.6 / 16.7 |
| FDA$^{(3)}$ | 73.9 / 58.0 | 77.4 / 72.4 | 58.3 / 42.1 | 76.9 / 72.8 | 70.7 / 50.4 | 57.0 / 42.3 | 55.2 / 11.1 | 57.4 / 10.7 | 77.9 / 22.1 |
| FDA$^{(4)}$ | 76.1 / 60.7 | 79.4 / 74.9 | 56.0 / 41.5 | 79.1 / 75.3 | 73.0 / 53.2 | 55.0 / 42.1 | 58.1 / 12.1 | 60.4 / 11.4 | 80.1 / 22.5 |
| FDA$^{(5)}$ | 76.5 / 61.9 | 79.7 / 75.4 | **59.2 / 45.5** | 79.4 / 75.9 | 73.5 / 54.0 | **58.4 / 45.5** | 58.8 / 12.2 | 60.7 / 11.7 | 80.5 / 22.4 |
| FDA$^{(1)}$+xent | 70.2 / 50.1 | 75.6 / 69.4 | 31.7 / 17.2 | 72.0 / 65.8 | 69.0 / 45.6 | 28.5 / 16.0 | 55.0 / 7.9 | 58.3 / 9.0 | 76.1 / 16.4 |
| FDA$^{(2)}$+xent | 74.5 / 58.3 | 79.8 / 75.2 | 43.9 / 29.8 | 77.2 / 73.0 | 73.3 / 53.0 | 42.4 / 30.7 | **60.4** / 10.5 | **62.1** / 10.7 | **81.2** / 19.4 |
| FDA$^{(3)}$+xent | 74.3 / 59.3 | 79.6 / 76.1 | 47.0 / 33.3 | 78.7 / 75.4 | 71.7 / 52.4 | 46.1 / 35.0 | 55.6 / 11.5 | 57.9 / 11.4 | 78.3 / 23.8 |
| FDA$^{(4)}$+xent | 76.5 / 62.2 | 81.3 / 78.0 | 45.9 / 32.7 | 80.8 / 77.7 | 74.1 / 55.3 | 44.7 / 34.0 | 58.7 / 12.6 | 61.0 / 12.1 | 80.6 / **24.3** |
| FDA$^{(5)}$+xent | **77.0 / 63.1** | **81.8 / 78.5** | 48.8 / 36.3 | **80.9 / 77.8** | **74.4 / 56.2** | 47.8 / 37.5 | 59.3 / **12.6** | 61.3 / **12.2** | 81.1 / 23.9 |

target model case, $FDA^{(5)}$+*xent* induces 77% error / 63% tSuc when transferred from the RN50-A whitebox, which is an improvement over *TMIM* of about 40% error / 51% tSuc. When compared to the $FDA^{(1)}$ method, our multi-layer extensions lead to improvements of over 9% error / 17% tSuc. Similar results are shown in RN50-B→DN121-A.

**Scenario 2.** Next, we analyze the case in which the blackbox model is trained on a subset of the whitebox model's training dataset, i.e., RN50-Full→DN121-A/B. Different from the previous experiment, the complexity of the whitebox and blackbox tasks is significantly different (1,000 class versus 15 class classification), where the whitebox model's task is more challenging. Remarkably, despite such a gap in the task and dataset complexity, the transfer results from the feature distribution attacks are quite high. In both cases, the error is near 60% and tSuc is over 45%. This is as compared to the near 16% error / 5% tSuc from the *TMIM+SGM* attack. Again, we also see large improvements using multi-intermediate-layer attacks. Interestingly, the +*xent* component is not helpful in this situation. This is likely because the decision boundary structure of the two models is sufficiently different that it is no longer relevant for generating attacks.

**Scenario 3.** In the last case, the whitebox model is trained on a subset of the blackbox model's dataset, which is represented in the RN50-A/B/C→DN121-Full columns. Not surprisingly, the transfer rates of all methods are lower in this setting, likely because the complexity of the whitebox model's task is much simpler than the blackbox model's task. Thus, the level of detail and granularity of the features learned in the whitebox model is less. However, as with the other tests, all *FDA*-based methods significantly outperform the *TMIM* method, and the multi-layer upgrades significantly advance the performance over the vanilla $FDA^{(1)}$. An interesting takeaway is that the size of the whitebox's subset matters greatly. Even as we look from RN50-A/B to RN50-C as whitebox models, the performance increases by over 20% error / 12% tSuc.

### 4.3  Query-based extension

A criticism of transfer-based blackbox attacks is that the success rate can be dependent on the whitebox-blackbox pair: certain combinations may have more limited transfer in some instances, as can be seen when MNv2 is the blackbox model in Table 1. In contrast, query-based blackbox attacks directly estimate the gradient of the blackbox model via repetitive querying during attack generation [35, 12, 3], and typically achieve higher success rates than transfer-based attacks. The principle drawback to query attacks, which does not afflict transfer methods, is that for each adversarial example the attacker may have to query the target model tens-of-thousands of times to properly estimate the gradient, which in some cases may not be feasible due to time and monetary constraints, or potential threat detection systems in the target model. Motivated by query efficiency, Cheng et al. [3] incorporate the adversarial direction from a transfer-based attack as a prior in the P-RGF attack. To show a potential way to extend our methods, in situations where limited queries to the blackbox model are acceptable, we investigate using $FDA^{(5)}$+*xent* and *TMIM* methods as *priors* in [3]. To integrate with P-RGF, we extend the transfer methods to perturb for 15 total iterations. In the first 10, the noise is solely optimized on the whitebox model (as usual). Only in the last 5 iterations do we use the transfer direction as the prior in the P-RGF estimator. We find warm-starting the transfer direction in this way to be effective. The tSuc results for several transfer scenarios, under different query budgets, for *TMIM* / $FDA^{(5)}$+*xent* priors, are shown in Table 4.

Table 4: tSuc of transfer-based attacks used as a prior with P-RGF (prior = *TMIM* / $FDA^{(5)}$+*xent*)

| # Queries \\ Target | Whitebox Model = RN50 | | | Whitebox Model = DN121 | | |
|---|---|---|---|---|---|---|
| | DN121 | VGG16bn | MNv2 | RN50 | VGG16bn | MNv2 |
| 0 | 3.0 / 57.9 | 1.5 / 43.5 | 0.1 / 22.9 | 2.3 / 50.1 | 1.3 / 48.0 | 1.0 / 24.4 |
| 100 | 5.5 / 70.5 | 3.3 / 60.3 | 2.1 / 35.5 | 4.8 / 63.0 | 3.1 / 65.8 | 2.0 / 37.1 |
| 500 | 7.5 / 81.2 | 6.2 / 77.3 | 4.2 / 56.8 | 6.6 / 79.7 | 5.8 / 86.2 | 3.9 / 65.8 |
| 1000 | 10.2 / 87.3 | 9.9 / 86.5 | 6.9 / 71.9 | 9.0 / 88.9 | 9.3 / 93.5 | 6.4 / 82.3 |
| 2000 | 14.9 / **92.9** | 16.9 / **93.4** | 12.3 / **85.5** | 13.0 / **94.7** | 16.0 / **97.3** | 11.9 / **93.2** |

In all transfer cases, we observe that the $FDA^{(5)}$+*xent* attack yields a significantly better prior than *TMIM*, which is aligned with the performance gaps observed in previous experiments. As the number of queries to the blackbox model increases, the attack success rate increases, and the margin of performance between the two priors grows to nearly 80% tSuc. With only 100 queries, the attack success rate of $FDA^{(5)}$+*xent* increases by over 10%, and with 2,000 queries the average tSuc rate is over 90%. This result motivates future work on how to improve the feature space attack framework through the allowance of a limited budget of queries to the target model during attack generation.

# 5   Conclusion

We introduce a feature space-based adversarial attack framework that allows for perturbations along the extracted feature hierarchy of a DNN image classifier to achieve state-of-the-art targeted blackbox attack transferability. In the "standard" blackbox transfer case, where the source and target model share ImageNet-1K as a training dataset, our methods outperform output-layer-based attacks by a factor of $10\times$ and existing feature-space methods by a factor of $2-3\times$. These performance gains are attributed to the inclusion of *multi-layer* information, which leads to significantly higher disruption in the feature spaces of both the whitebox and blackbox. In a set of three "strict" blackbox transfer scenarios, where the training dataset and label spaces of the whitebox and blackbox models differ, we show that our methods maintain similar performance margins over the baselines, even when there is no training data overlap or a 985-class discrepancy in the label space. Finally, we show an extension of the method for situations when it is acceptable to query the target model during attack generation.

## Broader Impact

Unlike many areas of research, the very name of "adversarial machine learning," with key terms such as "attacker" and "defender," may predispose readers to believing that such research automatically bears a negative societal impact. Conversely, much of the research in this area, including this work, is in pursuit of more positive goals: vulnerability awareness, robustness, and understanding, as related to DNNs. Much as "white hat hacking" leads to more secure computer systems, by studying adversarial attacks, we are working to find potential exploits and vulnerabilities that may be used by a malicious party in the future. This is especially important as deep learning technologies are adopted into real-world applications. The goal would then be to make such deployed DNNs robust against known vulnerabilities so the predictions can be interpreted with confidence. We also study adversarial examples and attacking techniques to enhance our practical understanding of how DNNs make decisions, as we are examining the behavior of models near and around the decision boundaries, which are formed after a long and difficult-to-interpret training process.

In the context of society at large, a positive impact that this work may have is the development of more robust models in consumer products and applications. For example, in the space of self-driving cars, if the developers of the object detection and classification systems are aware of our method (along with many others) as a known vulnerability, the resulting system may not be fooled in the presence of a physical adversary who is trying to intentionally cause a malfunction of the system when the car is on the road. Another example is the creation of robust digital products and applications that rely on remotely-hosted models with secure/protected/unknown datasets (e.g., Google Cloud Vision). Previously, transfer attacks were primarily discussed in the context of the source and target model being trained on the same dataset. Therefore, a company/institution confident in their training dataset being under wraps may be lulled into a false sense of security with regards to vulnerability to transfer attacks on their system. The designers may then assume that throttling user query rate to hamper query-based adversaries is sufficient for adversarial threat mitigation. However, our work shows that a feature-based adversary can transfer even in cross-distribution settings, where the attacker needs not have access to the exact training data or label space of the proprietary model. Since transfer attacks have a very small query "profile," and do not require repetitive querying, such threat mitigation systems may easily be circumvented by the attack we have proposed. Informed with the renewed capabilities of a transfer adversary, designers of such remote and proprietary models may now consider other ways to improve the robustness of their models, resulting in an overall positive impact on the quality and safety of their machine learning systems.

There are also some potential negative consequences of the work on society, especially if the exposed adversarial vulnerabilities are not addressed in the design of future systems. Given trade-offs that system designers may have to make, such as accuracy/generalization vs. robustness, training time vs. robustness, development time vs. deadlines, it may be tempting to develop non-robust models for high accuracy, low training time, and low development time. However, if this is done, and the model is deployed without proper consideration of adversarial vulnerabilities including (but not limited to) our method, it may result in the crash of a self-driving car, a weapon passing through an X-ray baggage scanner, poisoned data samples being classified (and potentially re-trained on) by a remote system, etc. Overall, although there is potential for both positive and negative consequences, we believe the best use for our work is to be used in the development stages of a system to ultimately create more robust and safe models with interfaces to society.

## Acknowledgments and Disclosure of Funding

This research was supported by the Air Force Research Laboratory (AFRL) under grant FA8750-18-2-0057 and the Office of Naval Research (ONR) under grant N00014-18-1-2871. Kevin J Liang was supported by the E Bayard Halsted Scholarship.

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
