[Supplementary Material]

# Appendix

## A. Layer decoding tables

| Layer # | DN121 Layers | Sequence |
|---|---|---|
| 16 | 6_12_24_16 | |
| 15 | 6_12_24_12 | |
| 14 | 6_12_24_8 | 4 |
| 13 | 6_12_24_2 | |
| 12 | 6_12_24 | |
| 11 | 6_12_22 | |
| 10 | 6_12_18 | 1 |
| 9 | 6_12_14 | |
| 8 | 6_12_10 | 3 |
| 7 | 6_12_6 | |
| 6 | 6_12_2 | |
| 5 | 6_12 | 2 |
| 4 | 6_10 | 5 |
| 3 | 6_6 | |
| 2 | 6_2 | |
| 1 | 6_ | |
| Input Layer | | |

| Layer # | RN50 Layers | Sequence |
|---|---|---|
| 14 | 3_4_6_3 | |
| 13 | 3_4_6_2 | |
| 12 | 3_4_6_1 | |
| 11 | 3_4_6 | 4 |
| 10 | 3_4_5 | 1 |
| 9 | 3_4_4 | |
| 8 | 3_4_3 | |
| 7 | 3_4_2 | 3 |
| 6 | 3_4_1 | |
| 5 | 3_4 | 2 |
| 4 | 3_3 | 5 |
| 3 | 3_2 | |
| 2 | 3_1 | |
| 1 | 3_ | |
| Input Layer | | |

Figure 5: Layer notation of whitebox models and sequence in which layers get added to multi-intermediate-layer attacks.

Here we discuss the DNN layer notation used throughout the work. We use two whitebox models: ResNet50 (RN50) [9] and DenseNet121 (DN121) [10], which have been shown to be good sources for generating transferable adversarial examples.

The RN50 notation follows the implementation in `https://github.com/pytorch/vision/blob/master/torchvision/models/resnet.py`. RN50 has 4 layer groups, with {3, 4, 6, 3} Bottleneck blocks in each, respectively. Of the 16 possible layers we notate 14 of them in Figure 5 where "deeper" layers closer to the output of the model have higher layer numbers. This is the notation used in Figures 4 and 7.

The DN121 notation follows the implementation in `https://github.com/pytorch/vision/blob/master/torchvision/models/densenet.py`. DN121 also has 4 main layer groups, with {6, 12, 24, 16} layers in each, respectively. From the 58 possible layers, we sample 16 layers from across the depth as shown in Figure 5.

As an example of how the layers are used in the attacks, when generating an attack from a RN50 whitebox that uses layers 5 and 10, this means that we are "probing" the model to extract the feature map at the output of the {3,4} and {3,4,5} layers. The "Sequence" column is the order in which the 5 attacking layers get added to the multi-layer attack, as found by greedy optimization. See Appendix B for more information about this process.

## B. Additional experimental setup details

**Auxiliary models.** Using pseudo-PyTorch notation, we use the following architecture for the auxiliary models: `[Conv(#kernels=128, kernel_size=3, stride=1, pad=1), ReLU, MaxPool(2,2), Conv(#kernels=128, kernel_size=3, stride=1, pad=1), ReLU, MaxPool(2,2), Dropout(p=0.3), linear(inputs=?, outputs=1)]`. Note that the number of nodes in the linear layer depends on the spatial size of the input feature map. We use this exact architecture regardless of layer and feature map size, for simplicity reasons. Customizing the auxiliary architecture to each layer in the whitebox model may result in better modeling of the feature distributions at each layer, but we find this architecture to perform well enough.

Each auxiliary model is trained for 8,000 iterations with `batch_size=64`, `momentum=0.9`, `weight_decay=5e-4`, `initial_learning_rate=0.001`, and a single learning rate decay step at the $6,000^{th}$ iteration to 0.0001. For each batch, we use a weighted sampling scheme to equalize the counts of positive and negative training samples. Note, the training of each auxiliary model can be done in a massively parallel fashion.

**Greedy layer optimization.** Selection of the whitebox model layers to target with $FDA^{(N)}$ is a combinatorial problem. Rather than try out every possible combination, we use a greedy optimization strategy to find the best layers in Section 4, which we elaborate on here. While this greedy approach may not find the absolute optimum set of layers, we find that it still performs quite well, with each added layer delivering significant gains in attack performance.

Note, in [14] the "optimal" attacking layer is found via sweeping across possible layers and using the layer that empirically has the best tSuc rate. We adopt a similar scheme here. We only optimize to find the attack layers once. When finding the attack layers for the RN50 whitebox model we use the DN121 as a blackbox, and when finding the attack layers for DN121 we use the RN50 as a blackbox. Using a random held-out set of 200 source images from the ImageNet validation set, we generate attacks from each layer to find the best single layer to attack from. To find the best two layers to attack from, we start with the best single layer and sweep the second layer to find the best complement to the first layer. This process of using the best previous layers and finding the most complementary new layer continues until we find the best 5 layers, where we notice a performance saturation. The sequence of how the attacking layers get added is shown in the "Sequence" columns of Figure 5. For example, for the $FDA^{(3)}$ attack on a RN50 whitebox, the layers used in the attack are {3,4,5}, {3,4}, and {3,4,2}. The hyper-parameters in the objective functions are found via line search. For the RN50 whitebox, $\eta = 1e - 5$ and $\gamma = 1e - 4$. For the DN121 whitebox, $\eta = 1e - 6$ and $\gamma = 1e - 4$. For simplicity, in a multi-intermediate-layer attack all $N$ layers are equally weighted, so $\lambda_\ell = 1/N$.

**Baseline attacks.** The setup and configurations of the baseline attacks largely mirror the original papers. Since the results in the *TMIM+SGM* [36] paper are all tuned for un-targeted attacks, we do a similar line search on a held out dataset to find $\text{decay} = 0.5$ for RN50 and $\text{decay} = 0.4$ for DN121. For the ensemble attack [17], all models are weighted equally, and we include momentum in the optimization (although momentum is not discussed in the original paper) because it empirically improves the results.

All of the attacks, including new ones and the baselines use $L_\infty$ $\epsilon = 16/255$, $\text{step\_size} = 2/255$ and $\text{perturb\_iters} = 10$.

# C. Full ensemble comparison results

Figure 6: Comparison of FDA methods to ensemble attacks.

Figure 6 shows an extension of the results in Figure 2 to include all of the transfer scenarios considered in Section 4.1.1. The primary result is that our FDA methods, and particularly $FDA^{(N)}+xent$, outperforms the ensemble methods in all of these transfer scenarios in both error and tSuc. The critical detail is that in these plots, our method is still crafting noise using the feature space of a *single* source model, where the ensemble method is using output layer information from multiple source models of varying architectures and depths. Since FDA and ensemble are seemingly orthogonal methods, we leave it to future work to explore their combination.

## D. Extended analysis of disruption

Figure 7: Disruption of features caused by transfer attacks.

Here we extend the disruption results from Section 4.1.4 to include both RN50→DN121 and DN121→RN50. The disruption caused by each attack is measured over 5,000 adversarial examples from each method. Adhering to the setup in Section 4, all source (clean) samples are correctly classified by both the whitebox and blackbox. The results shown here echo the conclusions drawn in Section 4.1.4. The main conclusions are as follows:

- The *TMIM* attack [5], which specifically leverages the output layer of the whitebox model, is only capable of causing high disruption towards the output of the whitebox model. In both the early layers of the whitebox and all throughout the blackbox model, *TMIM* causes very little disruption w.r.t. the target class.

- The baseline $FDA^{(1)}$ method [14] causes significantly more disruption in the intermediate feature spaces of both the whitebox and blackbox but has two undesirable behaviors. First, the intermediate disruption is significantly higher in the whitebox than in the blackbox, even for the layers that were not considered in the attack. Ideally, the intermediate space of the blackbox model would show higher disruption. Second, the disruption in the final few layers of both the whitebox and blackbox has a sharp decrease (as best seen in the RN50 whitebox and blackbox). This indicates that although the intermediate feature maps look to be the target class with nearly $100\%$ probability, that does not automatically induce classification as the target class.

- The primary effect of the *+xent* component is to create higher disruption in the final few layers, which directly addresses the weakness discussed in the previous bullet. This behavior is perhaps best illustrated by comparing the $FDA^{(1)}$ and $FDA^{(1)}$*+xent* results in the rightmost plot (RN50 as a blackbox). From early through middle layers, the disruption is about the same, and in the final 3 layers the *+xent* version has higher disruption. As well as being quite intuitive behavior, this is evidence for why the attack performs better.

- The notable effect of using multi-intermediate-layers is significantly more disruption in the intermediate space of both whitebox and blackbox models. Looking at the whitebox models, as we go from $FDA^{(1)}$ to $FDA^{(2)}$, the disruption is noticeably increased in the earlier layers. This is not surprising, as from Figure 5, the second attack layer to be added in both cases is a layer closer to the input layer. This trend of increased disruption in earlier layers is also observed in the blackbox models. As we move from $FDA^{(2)}$ to $FDA^{(4)}$, from Figure 5 we have added 2 layers that are both deeper in the model than the second layer added, so we do not see increased disruption in the earlier layers. However, we do see increased disruption in the later layers, as expected because the fourth layer added is the closest to the output of the whitebox model. This pattern of how including early layers in the attack set causes early layer disruption and including late layers in the attack set causes late layer disruption matches intuition and provides insight into how the attack works.

# E. Cross-distribution experiment

| RINet-A | | RINet-B | | RINet-C | |
|---|---|---|---|---|---|
| class name | components | class name | components | class name | components |
| fish | [1, 2, 389, 392, 394] | fish | [3, 391, 393, 395, 396] | bird | [10, 11, 12, 13, 14] |
| bird | [10, 13,16, 17, 18] | bird | [12, 14, 15, 19, 20] | turtle | [33, 34, 35, 36, 37] |
| lizard | [38, 40, 42, 45, 46] | lizard | [39, 41, 43, 44, 47] | lizard | [42, 43, 44, 45, 46] |
| snake | [52, 55, 57, 61, 67] | snake | [53, 60, 62, 64, 68] | snake | [60, 61, 62, 63, 64] |
| spider | [72, 75, 76] | spider | [73, 74, 77] | spider | [72, 73, 74, 75, 76] |
| dog | [161, 166, 179, 231, 249] | dog | [162, 167, 180, 232, 250] | crab | [118, 119, 120, 121, 122] |
| cat | [281, 284, 287, 288, 293] | cat | [282, 285, 289, 290, 292] | dog | [205, 206, 207, 208, 209] |
| insect | [302, 304, 308, 311, 313] | insect | [303, 305, 310, 312, 315] | cat | [281, 282, 283, 284, 285] |
| boat | [403, 472, 554, 814, 833] | boat | [510, 625, 628, 724, 871] | bigcat | [289, 290, 291, 292, 293] |
| car-truck | [436, 468, 609, 654, 817] | car-truck | [511, 627, 656, 705, 717] | beetle | [302, 303, 304, 305, 306] |
| turtle | [33, 34, 35, 36, 37] | crab | [118, 119, 120, 121, 125] | butterfly | [322, 323, 324, 325, 326] |
| big-game | [347, 348, 349, 350, 351] | small-game | [356, 357, 359, 360, 361] | monkey | [371, 372, 373, 374, 375] |
| glassware | [572, 737, 898, 901, 907] | musical-instrument | [402, 420, 486, 546, 594] | fish | [393,394, 395, 396, 397] |
| train | [466, 547, 565, 820, 829] | computer | [508, 527, 590, 620, 664] | fungus | [992, 993, 994, 995, 996] |
| fungus | [992, 993, 994, 995, 996] | fruit | [948, 949, 950, 951, 954] | musical-instrument | [402, 420, 486, 546, 594] |
| | | | | sportsball | [429, 430, 768, 805, 890] |
| | | | | car-truck | [609, 656, 717, 734, 817] |
| | | | | train | [466, 547, 565, 820, 829] |
| | | | | clothing | [474, 617, 834, 841, 869] |
| | | | | boat | [403, 510, 554, 625, 628] |

Figure 8: Class splits of Restricted-ImageNet subsets.

To evaluate the effect of differences in the training data distributions of the source and target model we propose three splits of the ImageNet-1k dataset [4], which we call RestrictedImageNet-A/B/C (RINet-A/B/C). To split the classes, we leverage the WordNet [20] hierarchical structure of the dataset such that each class in a RINet set is a superclass category composed of multiple ImageNet-1K classes, noted in Figure 8 as "components." For example, the "fish" class of RINet-A (both the train and val parts) is the aggregation of ImageNet-1k classes: [1:'tench', 2:'goldfish', 389:'barracouta', 392:'rock-beauty', 394:'sturgeon']. See `https://gist.github.com/yrevar/942d3a0ac09ec9e5eb3a` for the number to category translations.

**RINet-A&B.** One of the key experiments in this work is to measure the transfer performance when the source and target models are trained on similar categories, but have no training data overlap. This represents a more realistic attacking scenario when the adversary is aware of what classes the target model is trained on (or at least most of them) but must collect their own data as they do not have access to the target model's training dataset. To simulate this scenario, we create RINet-A and RINet-B, which each have 15 classes, of which 10 are shared between the two and 5 are unique, and have *zero* training data overlap.

To perform our experiments, we trained a RN50 and DN121 model on each of the splits using code from `https://github.com/pytorch/examples/blob/master/imagenet/main.py`. The accuracy of each model is shown in Figure 9. Since it is not meaningful to test an RINet-A model on the unique classes of RINet-B, we show the test accuracy on the

| | Accuracy | | | |
|---|---|---|---|---|
| Model | A-Test-full | A-Test-shared | B-Test-full | B-Test-shared |
| RN50-A | 0.9564 | 0.9546 | * | 0.8146 |
| RN50-B | * | 0.8387 | 0.9435 | 0.935 |
| DN121-A | 0.966 | 0.9658 | * | 0.8467 |
| DN121-B | * | 0.8788 | 0.9616 | 0.955 |

Figure 9: Standard accuracy of RINet models.

"full" and "shared" classes only when appropriate. For the models trained and tested on the same splits, the accuracy is about 95%. When the models are tested on the shared class data of the other split, the test accuracy drops to about 81% − 87%, which is not surprising given the difference of ImageNet classes used.

**RINet-C.** The other dataset considered is RINet-C. It has very similar construction to A & B in that each class is an aggregation of five ImageNet-1K classes. However, RINet-C has 20 classes which makes it a bigger subset than RINet-A/B. The purpose of including RINet-C is to measure how important the size of the subset is when attacking in the RN50-A/B/C→DN121-FullINet and RN50-FullINet→DN121-A/B/C scenarios. We train RN50 and DN121 RINet-C models in the same way as we do the RINet-A/B models. The test accuracy of RN50-C is 95.2% and DN121-C is 96.3%.

**Cross-distribution attack settings.** Lastly, we discuss how we setup and carry out the attacks in the cross-distribution experiments. Here, we assume the auxiliary models have been trained for RN50-A/B/C/FullINet (training details in Appendix B).

- *Constants across all tests:* The attacking parameters are still $L_\infty \epsilon = 16/255$, $\text{step\_size} = 2/255$, and $perturb\_iters = 10$.

- *RN50-A/B→DN121-A/B (Scenario 1):* Since the source task is significantly different than Full-ImageNet, we re-search for the best 5 attacking layers for the RN50-A/B models. We use the same greedy search technique described in Appendix B only on the RN50-A→DN121-A transfer scenario. In the notation of Figure 5, the sequence of adding attacks layers in the *FDA$^{(N)}$+xent* framework is: `1:[3,4,6,1]`, `2:[3,4,6]`, `3:[3,4,6,2]`, `4:[3,4,5]`, `5:[3,4,4]`. We also find that increasing the weight of the cross-entropy term to $\gamma = 1e-3$ helps. Other attacking parameters stay the same: $\eta = 1e-5, \lambda_\ell = 1/N$. When attacking, we only consider source samples from the shared-classes that are correctly classified by both the source and target models. We then target each of the other 9 shared classes individually and do this for all source samples in the whitebox model's validation dataset. Given the classes are shared, the computation of error and tSuc is straightforward.

- *RN50-A/B/C→DN121-FullINet (Scenario 3):* In these tests, we use the exact layer-set and hyper-parameters used in Case 1. The only tricky part is how to define attack success because the class structures are significantly different. Here, since the whitebox model is RN50-A/B/C, we work in the label space of RINet-A/B/C. In the attack loop, for each source sample in the whitebox model's validation set that is correctly classified by both the whitebox and blackbox, we target each of the other 14 (or 19 in the case of RINet-C) classes individually. An attack is successful if the blackbox model's top-1 prediction is one of the components that makes up the RINet target class. For example, if we use RN50-A as the whitebox model, for the target label "spider" we count a tSuc iff the DN121-FullINet model's prediction is in $[72, 75, 76]$. This is also how we check that the blackbox model is initially correct for each source sample. Note, this is a somewhat restrictive definition of tSuc. If we attack with the target label "dog," we only count a tSuc if the blackbox model's prediction is a component that made up the RINet's dog class. However, ImageNet-1K has over 150 "dog" sub-classes but tSuc only accounts for 5 of these.

- *RN50-FullINet→DN121-A/B/C (Scenario 2):* In these evaluations, we use the attack configurations, layer set, and hyper-parameters from the main ImageNet transfer tests (i.e., we do not tune for this particular test). As in Case 2, since the label spaces of the whitebox and blackbox models are different, we must handle the conversion as to measure attack success. For each source image in the ImageNet-1K validation set, we first check that it is a component of the target model's dataset, then we check that it is correctly classified by both the whitebox and blackbox using their respective label spaces. Using images that passed both these checks, we then target each of the other 14 (or 19) classes in the RINet label space by randomly choosing a Full-ImageNet target label from the component set of the target RINet class. For example, if we want to target "bird" on the RN50-B blackbox model, we would randomly select a label from the component set $[12, 14, 15, 19, 20]$ and use that as the target label in the Full-ImageNet space. With this, computing error and tSuc is straightforward.

Lastly, we would like to emphasize that we did not re-tune all of the hyper-parameters and layer sets for each individual transfer. We mention this to illustrate that the method and results are not ultra-sensitive to these parameters (within reason). However, it would not be surprising if the transfer results improve if we tuned the hyper-parameters for each situation.

## F. Perturbed Samples

Figure 10: Samples of adversarial examples generated with the *TMIM*, *FDA*$^{(1)}$, and *FDA*$^{(5)}$+*xent* attacks using $\epsilon = 16/255$.