[Reviews · NeurIPS 2020]

Review 1

Summary and Contributions: - This paper proposes a black-box transfer-based adversarial attack method, which utilizes the intermediate features in a white-box model. The proposed method uses multiple layers for a stronger attack. - The proposed method has two components: (1) cross-entropy terms for enhancing targeted attacks, and (2) aggregation of multiple layers. Both components improve attack performance (in terms of error / tSuc). - This paper provides extensive experimental results, including cross-distribution scenarios.

Strengths: - The proposed method is simple yet effective in various black-box attack scenarios. - This paper considers more realistic attack scenarios, e.g., cross-distribution.

Weaknesses: - The first major concern is the limited methodological contribution compared to FDA. The proposed method just aggregates (i.e., sum) FDA objectives of multiple layers and adding the cross-entropy term like other attack methods; in other words, these approaches are straightforward. Although the improvements of the proposed method are meaningful, it is not surprising or interesting results. - Secondly, the comparision between TMIM/SGM and FDA-based frameworks seems to be unfair. TMIM/SGM methods do not use the training data for the white-box model while FDA-based frameworks use the data for training auxiliary functions g. In my opinion, access to only pre-trained white-box models largely differs from that to whole training data, and thus the latter uses more knowledge than the former. So the improvements over baselines seem to be somewhat overclaimed, especially when the white-box and black-box models are trained on the same dataset. If using the intermediate features is crucial in adversarial attack, then how to utilize the features without the training data? The authors partially cover this issue in "cross-distribution" scenarios (Section 4.2), but in that case the source's and target's label spaces are largely overlapped. I think a harder case should be considered; for example, all labels are exclusive, or the number of available training samples is small. - Is the greedy layer optimization important? How about selecting layers heuristically, for example, the feature maps right before the pooling layers? ========= I generally agree with the author's response about my concerns.

Correctness: Yes.

Clarity: This paper is generally well-written. The followings are some types or suggestions: - Clarification is required for general readers who are not familiar with adversarial attack. In Equation (1)-(3), using g_{l,y}(f_l(x+delta)) instead of p(y|f_l(x+delta)) would be better for understanding because the latter can be considered as the softmax output of the model f. - typo: Line 18, "the the". - typo: Line 52, "i.e." -> "i.e.,"

Relation to Prior Work: While I'm not familiar with the adversarial attack literature, I think this paper discusses prior works enough.

Reproducibility: Yes

Additional Feedback:


Review 2

Summary and Contributions: The paper proposes a targeted transfer-based blackbox attack by allowing perturbations in the intermediate layers. As it is building on an already existing framework (FDA), the method might be lacking novelty, however, the empirical evaluation is quite extensive.

Strengths: The paper is well written and the experiment section seems thorough.

Weaknesses: 1. It might be worth mentioning a connection between adversarial examples and counterfactuals. Can the proposed framework be used in interpreting/explaining models predicitons or behaviour? 2. Can you include some visuals of the generated adversarial examples? 3. From equation 3, we see that each layer contributes differently. Is there any intuition on petrurbations from which layers might have a bigger impact (more specifically are the deeper layers more important)? 3. Would it be possibe to have a human sample evaluation for the attack? 4. Could the author give an opinion if the sucess of the the proposed attack can be explained by "robust vs not-robust features" (Adversarial Examples Are Not Bugs, They Are Features https://arxiv.org/abs/1905.02175) 5. Will the code be made available?

Correctness: To the best of my knowledge the method seems correct.

Clarity: Yes.

Relation to Prior Work: Discussion on other existin feature space attackes could be included: - "Towards Feature Space Adversarial Attack" (https://arxiv.org/abs/2004.12385) - "Constructing Unrestricted Adversarial Examples with Generative Models" (https://arxiv.org/pdf/1805.07894.pdf)

Reproducibility: No

Additional Feedback: ------------------------------------------------------------ Update after Author Feedback and Discussion ------------------------------------------------------------ Thank you to the authors for their detailed feedback. My questions and concerns have been mostly addressed/answered so I raise my score. 


Review 3

Summary and Contributions: This paper discusses an approach for blackbox transfer-based adversarial attack for DNNs. The approach is a straight-forward extension of a prior work based on layer-wise perturbation of feature maps of a white box model to incorporation of multiple layers. The authors show that the proposed extension results in significantly better results for target attack accuracy on a number of combinations of source-target models and a variety of conditions in terms of overlap between classes and trining data.

Strengths: The results obtained by the authors on various combinations of white box and black box models for transfer of adversarial attack and very impressive. Furthermore, the improvements are quite consistent even in the extended study where the training data and labels are don't match for the two models. The paper is clearly written to explain the contributions. The experiments are pretty detailed cover a large number of practical conditions.

Weaknesses: Some of the analysis presented in Section 4.1.4 of multi-intermediate-layer is not clear. For example, which layers were used to perturb the feature-maps in the FDA(1) and other multi-layer models isn't shown. Will these plots change depending upon the what layers where chosen by the perturbation algorithm or it is indecent on the chosen layers? I also couldn't find any implication on the time complexity involved in extending the attack generation process to multiple layers. Although the authors do talk about such costs in 4.3 for query based extension but it will be good to comment on their own method as well compared to simple FDA.

Correctness: This is mostly an empirical paper and the experiments performed look fine to me.

Clarity: The paper is well written.

Relation to Prior Work: Yes.

Reproducibility: Yes

Additional Feedback: I have read the authors' rebuttal and I am fine with that.

[Author Response · NeurIPS 2020]

**Reviewer 1: (1).** We perceive the main contribution of the existing FDA to be a proof-of-concept that attacking from intermediate feature space using learned feature distributions works at the scale of ImageNet. The main results focus on finding the single best layer to transfer from, and analyzing why that layer may be "optimal." Importantly, we found no discussion of possible methods/benefits for extending to the multi-layer setting. Although our methods may appear "straightforward" from simply looking at the equations, we maintain that the multi-layer extension is a significant and important conceptual contribution beyond the first FDA paper. This importance is also demonstrated by a 2-3x performance gain over simple FDA and 10x gain over output-layer methods, which promotes practicality. We also maintain that the design/evaluation of the cross-distribution experiments, the distal transfers, and the query-based extension are significant contributions beyond the previous FDA work, which may be interesting to many other readers in the community. **(2).** We believe that assumed access to pretrained whitebox models is only practical when considering common benchmark datasets/tasks (e.g., ImageNet), and would caution against taking their existence for granted in general. Outside of common benchmarks, we argue that it is not easy to find pretrained models for any arbitrary task a target model may be trained on. So, we believe that when doing transfer attacks one does implicitly assume access to the same/similar training dataset of the target model, which reflects the generalized case where training a surrogate whitebox is necessary. Admittedly, this is a weakness of transfer attacks in general, and is a critical detail that is not readily discussed in contemporary works. For this reason, we design the cross-distribution experiments to take a significant step towards a more realistic setting. Finally, considering harder cross-distribution cases is intriguing, yet it is unclear what a targeted adversarial attack means if there is no label space overlap, e.g., how would one use a digit classifier to make a truck look like a goldfish? **(3).** The greedy layer optimization is important for maximizing performance. However, note from Figure 5 (appendix) the pattern in which the layers are sequentially "added" by the optimizer (from top to bottom: 4,1,3,2,5). We do not propose that this pattern will hold for all possible source models, but it may serve as a good rule of thumb for a heuristic approach. We have no reason to believe that the performance of a multi-layer attack would completely degrade if a sub-optimal layer set is chosen heuristically (to roughly match the patterns in Figure 5). We consider the direction of finding other optimizations for layer choice an important future work.

**Reviewer 2: (1).** Yes, we believe that the proposed framework may be used to interpret/explain models predictions or behaviors. For example, with feature distribution models placed throughout the depth of the classifier, we may interpret the evolution of a model's prediction by observing how the feature maps propagate through the layers. For the purposes of model and training analysis, a layer-wise disruption experiment (Figure 4) may help to align and analyze the features learned by two independently trained models. Or, using the highly transferable distal images, it may also be possible to analyze/visualize which features have been learned that are most shared. **(2).** Yes, we will include several visuals in the final submission. Note, we used the commonly utilized blackbox noise constraint of $L_\infty$ $\epsilon = 16/255$ (e.g. [5,6,36]), so the multi-layer FDA adversarial images qualitatively "look" very similar to the samples in the referenced works. **(3).** From eqn 3, you are correct, it is possible for all layers to contribute differently. However, on line 183 we mention that in this work all layers are weighted equally. We include $\lambda_\ell$ to express maximum flexibility in the multi-layer framework. In the future, a different optimization scheme may be able to change the relative layer weighting to improve the results. Also, consider Figure 5 (appendix), and observe the order in which layers get added by the greedy optimizer. Intuitively, the most impactful layers are added first. **(4).** Since the adversarial noise we use ($L_\infty$ $\epsilon = 16/255$) is quasi-imperceptible, it is unlikely to alter a human's ability to make a correct classification. Interestingly, (https://arxiv.org/pdf/1802.08195.pdf) discusses human sample evaluations for adversarial attacks using the $L_\infty$ constraint; however, they use much larger epsilon values, e.g., 32/255 and 40/255, on purpose to make the noise visible yet not entirely destructive. **(5).** The referenced paper posits that neural networks tend to learn both robust and non-robust features, and adversarial attacks tend to manipulate the non-robust features. Like other attacks, we believe our method may change some of the non-robust features to alter the classifier output. The difference in our work is that the signal we use to dictate the manipulation of the features is derived from the intermediate feature space rather than the output layer. We hypothesize that the reason our method has such significant transferability is because it better exploits the overlap of non-robust features shared by two distinct models. **(6).** Yes, we plan to make the code available upon request if/when the paper is published, and thank you, we shall consider the referenced related works.

**Reviewer 4: (1).** For the results in Section 4.1.4, we use the same attacking layers as used in the preceding results (e.g., Table 1). The decoding for this layer notation is shown in Figure 5 (appendix). For example, the FDA(1) attack uses layer 10 of the *whitebox* model, and FDA(4) uses layers 10, 5, 7, and 11. These attacking layers directly correspond to the layers on the x-axis of the whitebox subplot (but not the blackbox subplot). Any disruption caused in the blackbox model layers was not specifically optimized for by a FDA(N) attack, rather the disruption is a result of the perturbations crafted using the whitebox model's layers. We will be sure to clarify these points in the final version. **(2).** Actually, there is a close to negligible increase in time complexity over simple FDA for the attack generation process. The auxiliary classifiers are quite small (3 layer NNs, see Appendix B for more details), and training is embarrassingly parallel. During the actual attack, the additional computation for calculating the forward and backward pass through the auxiliary models is slight, regardless of if we are attacking with 1 layer (like simple FDA) or 5 layers. We remark that all of our experiments were done on a modest sized academic computation budget.

[Meta-Review · NeurIPS 2020]

After a discussion, the reviewers converged towards recommending to accept this submission. The reviewers were satisfied with the authors' response, and have updated their reviews accordingly.